# Radiomics Features in Predicting Human Papillomavirus Status in Oropharyngeal Squamous Cell Carcinoma: A Systematic Review, Quality Appraisal, and Meta-Analysis

**DOI:** 10.3390/diagnostics14070737

**Published:** 2024-03-29

**Authors:** Golnoosh Ansari, Mohammad Mirza-Aghazadeh-Attari, Kristine M. Mosier, Carole Fakhry, David M. Yousem

**Affiliations:** 1Department of Radiology, Northwestern Hospital, Northwestern School of Medicine, Chicago, IL 60611, USA; golnoosh.ansari@northwestern.edu; 2Division of Interventional Radiology, Department of Radiology and Radiological Sciences, Johns Hopkins School of Medicine, Baltimore, MD 21205, USA; 3Department of Radiology and Imaging Sciences, Indiana University School of Medicine, Indianapolis, IN 46202, USA; kmosier@iupui.edu; 4Department of Otolaryngology, Johns Hopkins School of Medicine, Baltimore, MD 21205, USA; cfakhry@jhmi.edu; 5Division of Neuroradiology, Department of Radiology and Radiological Sciences, Johns Hopkins School of Medicine, Baltimore, MD 21205, USA; dyousem1@jhu.edu

**Keywords:** radiomics, texture, head and neck squamous cell carcinoma, oropharyngeal squamous cell carcinoma, CT, MRI, cancer, human papillomavirus

## Abstract

We sought to determine the diagnostic accuracy of radiomics features in predicting HPV status in oropharyngeal squamous cell carcinoma (SCC) compared to routine paraclinical measures used in clinical practice. Twenty-six articles were included in the systematic review, and thirteen were used for the meta-analysis. The overall sensitivity of the included studies was 0.78, the overall specificity was 0.76, and the overall area under the ROC curve was 0.84. The diagnostic odds ratio (DOR) equaled 12 (8, 17). Subgroup analysis showed no significant difference between radiomics features extracted from CT or MR images. Overall, the studies were of low quality in regard to radiomics quality score, although most had a low risk of bias based on the QUADAS-2 tool. Radiomics features showed good overall sensitivity and specificity in determining HPV status in OPSCC, though the low quality of the included studies poses problems for generalizability.

## 1. Introduction

Head and neck squamous cell carcinoma (HNSCC) is the seventh most common cancer globally, accounting for more than 300,000 annual deaths [1]. Alarmingly, it is projected that there will be a 30% increase in the overall incidence of HNSCC globally, with both developed and developing nations experiencing a growth in case numbers. This trend is driven by human papillomavirus (HPV), especially in North America and Europe [2,3], although there are other risk factors involved in the development of HNSCC such as alcohol consumption, tobacco products, blood group type, and more [4,5]. Even today, it is estimated that the frequency of HPV-positive oropharyngeal (OP) SCC is 2.5 times that of HPV-negative OPSCC [6]. Furthermore, although HPV-positive and -negative HNSCC have conventionally been classified as a single clinical diagnosis and represented as a single entity in the ICD classification [7], they show different clinical profiles and have recently been viewed as two distinct clinical entities by the National Comprehensive Cancer Network [8]. HPV-positive OPSCC usually has a higher 5-year overall, progression-free, and recurrence-free survival compared to HPV-negative OPSCC. It has also been shown that responses to conventional chemo-radiotherapy and immunotherapy are significantly different between the two groups [9,10]. Importantly, there are also critical differences between the two in tumor staging: the American Joint Committee on Cancer (AJCC) staging guidelines now offer two different T and N staging systems for OPSCC depending upon HPV status, essentially classifying them as two distinct clinical entities [11,12].

Conventionally, tissue samples have been used to document the presence of HPV in cancer cells, but recently, texture analysis features extracted from medical images have been proposed as a potential means to differentiate between HPV-positive and -negative OPSCC [13]. Radiomics features are mathematically extracted as high-throughput quantitative features that represent valuable information that is not readily/visibly appreciated by radiologists [14]. These features have the potential to substitute for tissue sampling as a form of virtual biopsy, as subtle nuances in the texture of a given lesion may be indicative of differences in microanatomy, which itself could be a reflection of different genomic alterations and pathologic processes in the cancer [15,16].

Currently, there is a lack of meta-analytic evidence regarding the possible role of radiomics features in determining HPV status in HNSCC [17]. Differences in the methodology of these studies, such as feature extraction, modality of choice, and issues regarding classification and validation, limit the generalizability of the results of isolated studies.

This investigation aims to determine the value of radiomics features extracted from medical images including CT, MRI, and ultrasound in determining HPV status in OPSCC and to determine the quality of radiomics methodologies used in the published literature.

## 2. Materials and Methods

### 2.1. Literature Search

Two researchers (G.A., M.M.A.A.) independently searched the PubMed, Scopus, Embase, and Web of Science repositories using the following combinations:

((HPV) AND (Radiomics)) AND (HNSCC), ((human papilloma) AND (Radiomics)) AND (HNSCC), ((papilloma) AND (Radiomics)) AND (head and neck), ((papilloma) AND (Radiomics)) AND (opscc), ((HPV) AND (Radiomics)) AND (opscc), ((papilloma) AND (texture)) AND (opscc), ((papilloma) AND (texture)) AND (hnscc), ((HPV) AND (texture)) AND (hnscc), (radiomics) AND (HPV), (radiomics) AND (human papillomavirus), (texture) AND (human papillomavirus).

Exclusion criteria consisted of studies that were not in English, conference papers, editorials, reports from society meetings or Kaggle challenges (open dataset challenges recruiting teams to train and validate predictive models or similar schemes) [18], abstracts, and review articles.

The same researchers were also responsible for screening the articles based on title and abstract to exclude irrelevant studies. After the suitable studies were identified, information regarding their methodology and numeric findings (namely the number of true positive, true negative, false positive, and false negative cases) were extracted. Only studies reporting complete metadata were included in the meta-analysis.

### 2.2. Quality Assessment

The radiomics quality score (RQS) and Quality Assessment of Diagnostic Accuracy Studies 2 (QUADAS-2) were used by the two researchers (G.A. and M.M.A.A.) to evaluate the methodological quality and risk of bias of the studies included in the systematic review. The two readers independently graded the studies and any case of disagreement was solved via discussion until consensus was achieved.

### 2.3. Statistical Analysis

Statistical analyses were performed with the MIDAS package Stata software, version 16.0, and the Meta-DiSc 2.0 web application. Statistical heterogeneity was assessed using the I2 value, providing an estimate of the percentage of variability among the included reports. I2 values of 0–25%, 25–50%, 50–75%, and >75% represented very low, low, medium, and high heterogeneity, respectively.

Pooled sensitivity, specificity, diagnostic odds ratio (DOR), positive and negative likelihood ratios (PLR and NLR), and area under curve (AUC), with corresponding 95% confidence intervals (CIs), were calculated. A forest plot and summary receiver operating characteristic (SROC) plot were generated for each of the modalities and in total. The pooling of studies and effect size were evaluated using a random-effects model, and sub-group analysis was performed to assess the diagnostic accuracy of radiomics features extracted from each imaging modality used.

Deek’s funnel plot was used to determine publication bias, and Fagan’s nomogram was used to determine clinical utility.

## 3. Results

### 3.1. Characteristics of the Included Studies

Twenty-six studies were included in the systematic review. Of these studies, only 14 studies containing 21 databases reported complete diagnostic results, which were included in the meta-analysis. One study was later excluded from the analysis due to reporting results obtained from patients with different types of HNPCC. A PRISMA flow chart of the study is presented in Figure 1. Table 1 Presents more information about the studies included in the study. OPSCC was by far the most common head and neck cancer studied, as 23 of the studies in the systematic review only included patients with OPSCC. Two studies included a variety of HNSCCs, and a single study included non-OPSCC HNSCCs. Only studies including OPSCC were included in the meta-analysis. Contrast-enhanced CT was the most prevalent modality used for texture feature extraction (sixteen studies), followed by MRI (eight studies using different combinations of sequences used for extraction of features), and PET-CT (two studies). The earliest study we were able to find was published in 2015, with 16 studies published between 2020 and 2023.

### 3.2. Methodological Quality

The QUADAS-2 tool was used to determine the methodologic qualities of the included manuscripts in the systematic review (Appendix A). Generally, most of the studies had a low risk of bias in patient selection, index testing, and timing. However, there was an undetermined risk of bias regarding the reference test, as different studies utilized different methods to determine if a lesion was HPV positive, including the detection of HPV DNA, HPV messenger RNA (mRNA), or the p16 protein. Polymerase chain reaction has inherent limitations in determining HPV infection, with the most important being the fact that PCR cannot differentiate low-risk strains of HPV from transcriptionally active ones, which are shown to be involved in the pathogenesis of OPSCC. radiomics quality score of the included articles is presented in Appendix A and Figure 1.

### 3.3. Publication Bias

Figure 2 depicts the Deek’s plot of asymmetry and the funnel plot of the study. There was no significant publication bias among the publications included in the meta-analysis.

### 3.4. Diagnostic Accuracy of Radiomics MRI

Twenty-one datasets pertaining to 14 studies were initially sought to be included in the meta-analysis. Of these studies, 13 studies incorporating 19 datasets exclusively included OPSCC cases, while a single study [29] did not determine the exact origin of the lesions included in their investigation. There were 1104 HPV-positive cancers and 748 HPV-negative lesions included. The pre-test probability of having the condition equaled 59%.

Figure 3 and Figure 4 depict the forest plot for sensitivity and specificity of radiomics features in determining HPV status in HNSCCs. The overall sensitivity of the included studies was 0.78 (0.74, 0.82), and the overall specificity was 0.76 (0.71, 0.81). The diagnostic odds ratio (DOR) equaled 12 (8, 17).

Subgroup analysis was performed based on modality, and the relative sensitivity level for MRI vs. CT comparison was 0.9 (sensitivity of 0.7, 0.8, respectively, 0.8–1.0, *p* = 0.27), and the relative specificity level for MRI vs. CT comparison was 1.0 (specificity of 0.79, 0.73, respectively, 0.9–1.1, *p* = 0.5) showed no significant difference between the two modalities. The total accuracy of datasets including MRI was 0.76, and the total accuracy of the studies including CT was also 0.76. Figure 5 presents the SROC curve of the studies based on the two modalities. It is evident that a lesser degree of variability was seen for results pertaining to datasets that used CT imaging.

Summary findings for a pre-test HPV-positive prevalence of 59% are presented in Figure 6.

### 3.5. Heterogeneity Assessment

The I2 statistic showed that heterogeneities for sensitivity and specificity were medium (I2 = 56.9% and 55.4%, respectively).

### 3.6. Clinical Utility

Figure 7 depicts the Fagan nomogram. Using a radiomics model generated on cross-sectional imaging would increase the post-test probability to 83% from 58%, with a positive likelihood ratio of 3.0 when the pre-test was positive. When the pre-test was negative, the post-test probability decreased to 29% with a negative likelihood ratio of 0.3.

## 4. Discussion

In the present review, we present meta-analytic evidence regarding the possible role of radiomics features extracted from CT and MR in determining the status of HPV in oropharyngeal squamous cell carcinoma. Our results show that the combined sensitivity and specificity of texture features equaled 78% and 76%, respectively, with studies using CT imaging showing less variability in their results. Interestingly, the following descriptive characteristics and their quantitative representative texture features were consistently observed to differentiate between HPV-positive and -negative lesions: HPV-positive lesions were more homogenous, spherical, and smaller, contained fewer clusters of areas with low HU (probably due to less tissue necrosis) [45], and showed lower ADC values.

Methodologically, most studies included patients from a single center for the development of their model, did not perform external validation, and acquired low radiomics quality scores. Furthermore, there are methodologic nuances that further limit the extent to which the results could be generalizable to different clinical settings: one crucial issue being the lack of utilization of Image Biomarker Standardization Initiative (IBSI)-approved extraction tools and IBSI-approved nomenclature for texture features of interest [46,47]. This is of particular importance since the next step in the implementation of radiomics to this population is the employment of established radiomics classifiers on external datasets and validation of competing classifiers, both of which could highly benefit from established IBSI performance benchmarks [29,48,49]. For example, some of the studies included in our meta-analysis utilized cases from an already established dataset on the Cancer Imaging Archive, which could serve as an opportunity for collaborative model development and external benchmarking of trained models [50,51]. Another potential factor that could limit the generalizability of the results included in the meta-analysis and introduce a means of heterogeneity is the differences in inclusion criteria of patients, especially the inclusion of patients in different stages of OPSCC. HPV-positive stage III-IV OPSCC lesions may have more histologic and anatomic similarity to HPV-negative lesions compared to lower-stage HPV-positive OPSCCs. This particular variation in patient inclusion may result in overly optimistic results in studies including a larger cohort of patients with earlier-stage OPSCC. This is of particular importance as most of the included studies in the meta-analysis did not report data regarding the stages of the included tumors, increasing their potential risk of bias in patient selection. Furthermore, of the 14 studies included in the meta-analysis, 13 exclusively focused on OPSCC, while a single study did not determine the exact origin of the lesions included, though the majority of their patient cohort consisted of patients with OPSCC (more than 75%) [41].

HPV is a well-established biomarker of prognosis. Routine HPV testing for oropharynx cancers is indeed recommended in NCCN guidelines, and clinical trials also incorporate HPV status with de-intensification trials focusing on HPV-positive patients [8]. Furthermore, the NCCN and updated American Joint Commission on Cancer guidelines consider them separate clinical conditions. This growing clinical importance of determining the status of HPV in HNSCC has led multi-disciplinary researchers to develop and validate surrogate markers that can readily determine high-risk infection [52]. One of the most promising developments has been the introduction of “NavDX^®^ liquid biopsy for the detection of circulating tumor-modified HPV DNA [53]. The lack of consensus regarding the definition of a clear cut-off for a positive immunohistochemical (IHC) test and concerns regarding varying sensitivity and specificity rates of this method arising from technical considerations have further necessitated a revisit of diagnostic approaches [54]. Other possible supplements to IHC have been the utilization of polymerase chain reaction (PCR) and in situ hybridization to detect HPV DNA [52,55,56]. However, these new methods may face barriers to feasibility and wide-scale application [13].

Since medical imaging of HNSCC and OPSCC is an essential part of staging the tumor and determining subsequent medical or surgical treatment plans, radiomics texture features offer a potential solution to characterizing the HPV status of these tumors. Our results show that the pooled sensitivity and specificity of radiomics features (77.2% and 76.3%) were below that of the currently established methods mentioned above, as IHC has shown a sensitivity and specificity close to 90% [57]. Though the current pieces of evidence do not suggest a substitutive role for radiomics features instead of routine para-clinical practices such as IHC, they provide insight regarding the additive value of radiomics features to these established diagnostic methods. Furthermore, artificial intelligence algorithms seem to be improving with each successive month, and, as more cases of HPV-positive and -negative cancers are accumulated, it is likely that radiomics features will become more reliable and accurate. This may be true not only for HPV status but also for prognostication and predicting response to treatment without regard to the HPV status of the lesion [58]. Future investigations should leverage the existing publicly available datasets and classifiers to develop, train, and externally validate new features [59]. These endeavors should also take into account clinical variables such as tumor grading and staging in developing predictive models and investigate the possible associations between characteristics such as blood types (recognized as risk factors for hypopharyngeal and oral cavity cancers) and clinical course of HNSCCs [60,61]. Future studies should also focus on increasing the interpretability of models by introducing feature maps and SHAP summary plots [62].

## 5. Conclusions

The present meta-analysis showed a combined sensitivity and specificity of 77.2% and 76%, respectively, for radiomics features in determining HPV status in OPSCC. The accuracy of radiomics features extracted from imaging is lower than that of the already established para-clinical IHC methods. However, since CT and MR imaging are essential in the work-up of most OPSCCs, collaborative learning models may uncover untapped potential for radiomics features in the determination of HPV status, prediction of treatment response, and other important factors in the diagnosis and therapy of OPSCC.

## Figures and Tables

**Figure 1 diagnostics-14-00737-f001:**
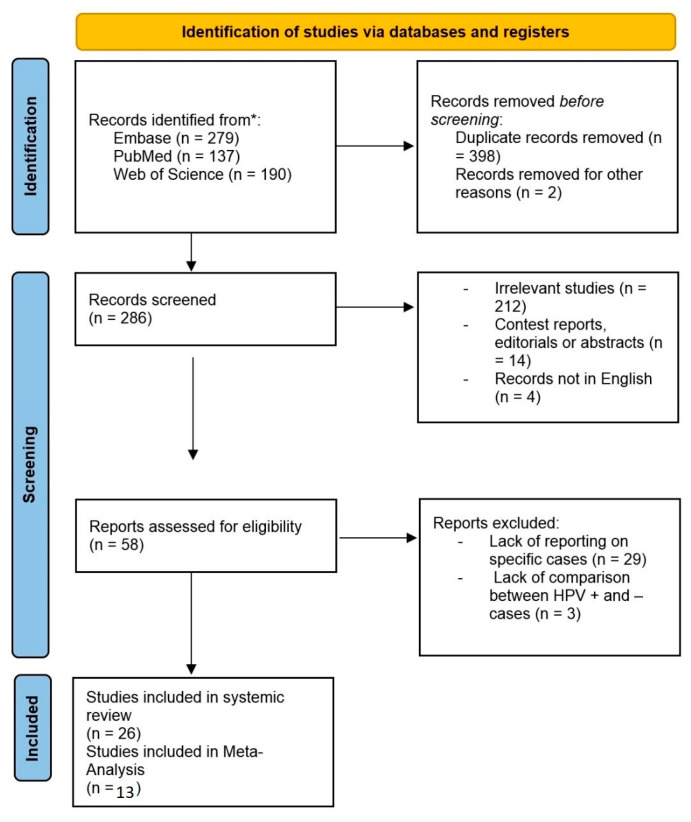
PRISMA diagram of the study, showing the included studies.

**Figure 2 diagnostics-14-00737-f002:**
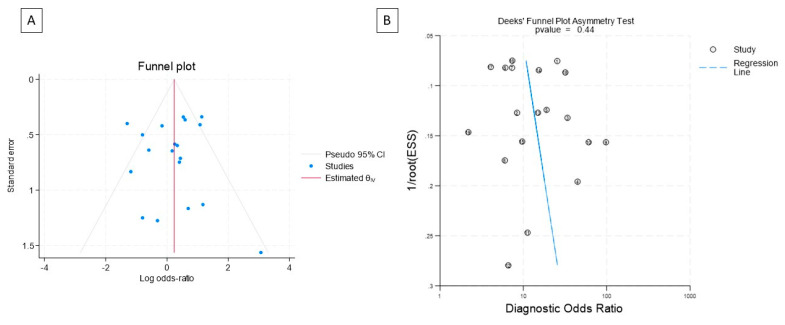
(**A**) Funnel plot with pseudo 95% confidence interval and (**B**) Deek’s funnel plot of asymmetry of the included studies. The two plots show that although there was some heterogeneity witnessed on the funnel plot, there was no significant publication bias in the included articles.

**Figure 3 diagnostics-14-00737-f003:**
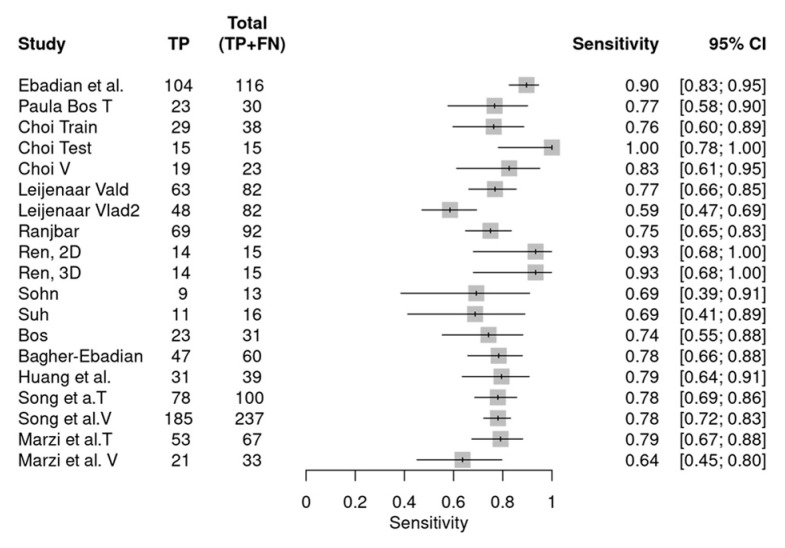
Forest plot showing the individual sensitivity of the included studies and their respective datasets in the meta-analysis [20,21,22,23,24,31,32,33,34,35,36,37,38].

**Figure 4 diagnostics-14-00737-f004:**
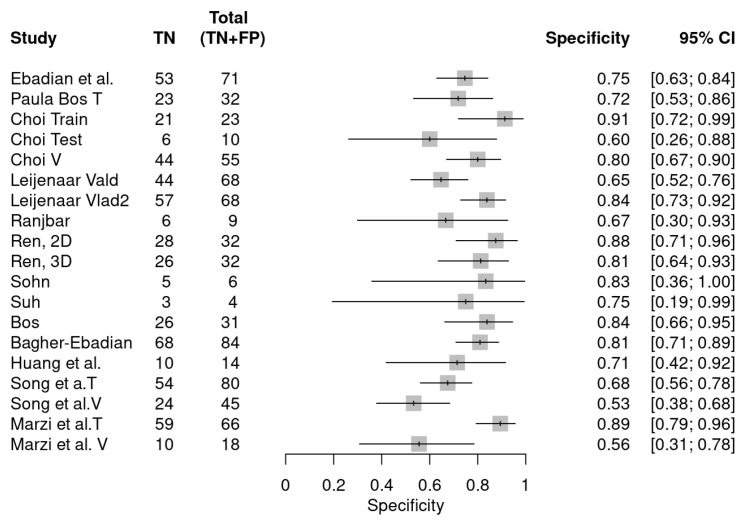
Forest plot showing the individual specificity of the included studies and their respective datasets in the meta-analysis [20,21,22,23,24,31,32,33,34,35,36,37,38].

**Figure 5 diagnostics-14-00737-f005:**
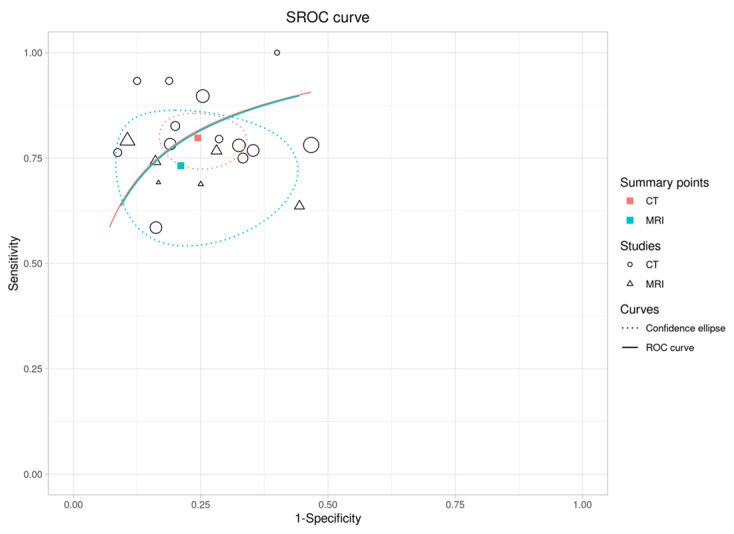
SROC curve of the studies included in the meta-analysis based on their utilized modality. The red line and the circles show studies using CT, and the green lines and triangles show studies using MRI. CT imaging showed a lesser degree of variation in diagnostic accuracy results compared to MR imaging.

**Figure 6 diagnostics-14-00737-f006:**
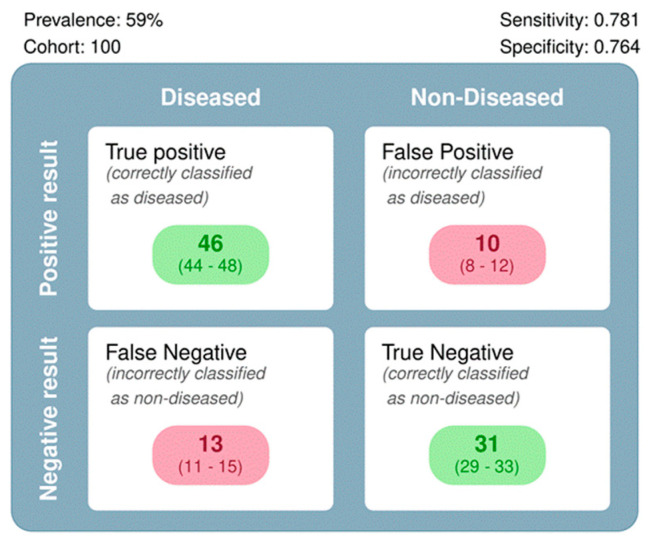
Summary findings of the meta-analysis, based on a pre-test prevalence of 58% and a hypothetical sample size of 100.

**Figure 7 diagnostics-14-00737-f007:**
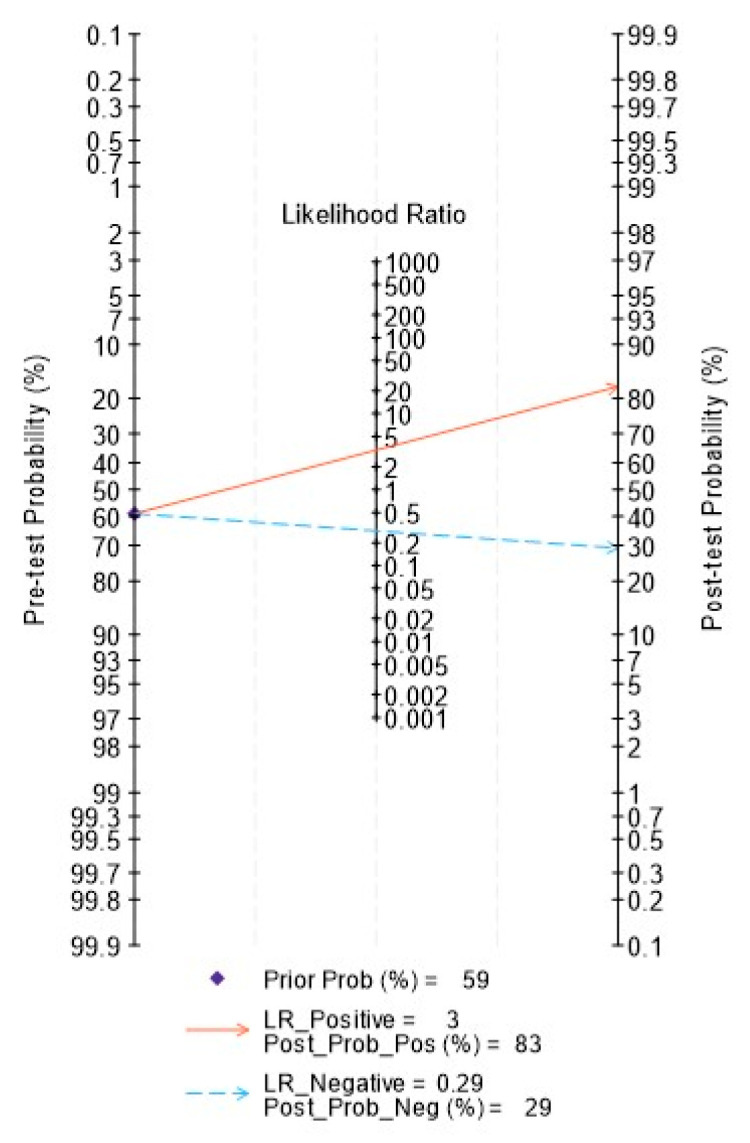
Fagan nomogram depicting the clinical utility of radiomics features in determining HPV status in the datasets included in the meta-analysis.

**Table 1 diagnostics-14-00737-t001:** Characteristics of the articles included in the systematic review. Bold articles are those that are also included in the meta-analysis. Bold feature extraction software are those adherent to IBSI nomenclature.

Country	Year	Study Name	Retro/Pro	Cancer	Modality	Number of Features Used for Model Development/Considered Significant	Extracted Features, Number Incorporated into Model	Name of Features Included in the Model (If Available/Applicable)	Technique Used for Feature Selection	Classification Method	Segmentation	Feature Extraction	Seg Software	Major Texture/Biologic Correlate (If Any)
The Netherlands	2022	Boot et al. [19]	Retrospective	Oropharyngeal squamous cell carcinoma	Pre-treatment T1	20	498	Run entropy, 10th percentile, dependence count energy, compactness, center of mass shift, minor axis length, asphericity, Zone distance non-uniformity GLDZM, flatness, high dependence, low gray level emphasis, coefficient of variation, and quartile coefficient	RFE–random forest, RFE–logistic regression	Random forest, logistic regression	Manual/semiautomatic	RaCat softw	Velocity AI 3.1	HPV-negative tumors are more heterogenous, while HPV-positive lesions are more spherical and have a clear defined margin
**The Netherlands**	**2022**	**Bos et al. [20]**	Retrospective	OPSCC	T1W MRI	20-9	1184	Not determined individually	ICC + Mann–Whitney U	Logistic regression prediction	Manual	**PyRadiomics**	3D slicer software	2D texture features could be used in lieu of 3D volume-of-interest features in differentiating HPV-negative and -positive lesions.
**United States**	**2022**	**Bagher-Ebadian [21]**	Retrospective	OPSCC	Contrast-enhanced computed tomography	14	192	Morphology, IBS, GLSZM, GLDZM, NGLDM, GLCM, GLRLM	Non-parametric tests, correlation analysis, LASSO	Logistic regression	Manual	ROdiomiX	ROdiomiX	Tumors of HPV+ patients are smaller and more spherical compared to the HPV- patients. Compared to HPV- patients, the absolute magnitude of HU of tumors in HPV+ patients is higher
Republic of Korea	2022	Park et al. [22]	Retrospective	OPSCC	T1 Gd-enhanced axial images	8	140	Skewness, maximum, 10 percentile, minimum surface volume ratio, coarseness, correlation, lmc2, contrast1	LASSO	Logistic regression model and a light gradient boosting machine (LightGBM)	Manual	**Pyradiomics**	3D slicer	In HPV-positive OPSCC, tumor boundaries were more clear and showed cystic changes in lymph node metastasis. It was also reported that the minimum apparent diffusion coefficient of HPV-positive OPSCC was remarkably low
China	2022	Wenbing et al. [23]	Retrospective	OPSCC	PET/CT	7	112	Un-named radiomics features selected based on saliency maps	AUC of each feature	Multivariate logistic regression model	Semi-manual	SERA; Ver 2.1	SERA; Ver 2.1	High saliency values were more frequently located at the remote region relative to the centroid for both PET and CT images, indicating the aggressiveness and special biological behavior of the tumor periphery
**Italy**	**2022**	**Marzi [24]**	Retrospective	OPSCC	MRI diffusion map	2	157	50th percentile of DPT, inverse difference Moment of ADCLN	Minimum redundancy maximum relevance (MRMR)	Naive Bayes classification	Manual	**IBEX**	3D slicer	HPV-positive tumors had a marked reduction in ADC values and a more leptokurtic and skewed right ADC histogram
**Republic of Korea**	**2021**	**Sohn [25]**	Retrospective	OPSCC	T1WI and fat-saturated	6	170	firstorder_Skewness, glcm_Imc1, ngtdm_Coarseness, firstorder_Skewness, ngtdm_Strength, shape_Flatness	LASSO	Based on coefficients of LASSO	Manual	**Pyradiomics 2.1.0**	3D slicer	N/A
United States	2021	Bagher-Ebadian [26]	Retrospective	OPSCC	CE-CT	15	192	Major axis length, GLSZ—zone size non-uniformity, root mean square intensity, minor axis length, GLDZM—small distance low gray level emphasis, NGLDM—high dependence high gray level emphasis, GLCM—information correlation, GLCM—correlation, MORPH—flatness, GLRLM—long run low gray level emphasis, GLDZM—large zone low gray level emphasis, GLSZM—large zone low gray level emphasis, NGLDM—dependence count variance, IH—maximum histogram intensity	LASSO	Filtered LASSO-GLM classifier	Manual	ROdiomiX	ROdiomiX	Intensity-based features (IVH, IH, IBS) became significantly different between HPV-negative and -positive groups in the post-frequency filtered datasets, suggesting a possible role of frequency analysis
**United States**	**2021**	**Song et al. [27]**	Retrospective	OPSCC	CE-CT	15	454	Kurtosis of diff-average, skewness of sum-average, median, λ = 3, θ = 0.1 rad, skewness, λ = 5, θ = 0.0 rad	MRMR	LASSO Score	Manual	In-house-developed	N/A	Higher Haralick correlation shows less pixel intensity disorders and decreased morphologic appearance heterogeneity for HPV-positive tumors compared with HPV-negative ones
**USA**	**2020**	**Ebadian et al. [28]**	Retrospective	OPSCC	Pre-treatment CE-CT	14	172	DOST- ENERGY-1, DOST- ENERGY-2, DOST- ENERGY-3, DOST- ENERGY-6, DOST- ENERGY-7, DOST- ENERGY-8, DOST- ENERGY-9, GLCM—contrast, GLCM—energy, GLCM—entropy, mean breadth 13, spherical disproportion	Levene KS test absolute biserial correlation	GLM	Manual	**MatLab**	MatLab	HPV+ tumors have a lower component density compared to the HPV– patients. HPV+ tumors have higher image contrast, with images being patchier than HPV-negative tumors. HPV+ tumors are more simple, symmetric, and more rounded
**Switzerland**	**2020**	**Bogowicz et al. [29]**	Retrospective	HNSCC (6 cohorts)	Pre-treatment CE-CT	10 centralized/12 distributed	981	HLH GLCM cluster shade, LHH GLCM cluster shade, HLH 10th percentile, HHL GLCM cluster shade, LLH NGLDM Idhge, LHH GLSZM sze, HLL GLDZM idlge, asphericirty, LLH GLSZM SZE, HLL GLSZM SZE, LHL GLSZM sze	ICC + the average linkage hierarchical clustering + univariate logistic regression	Multivariate logistic regression model	Manual	Z-Rad radiomics software	Z-Rad radiomics software	N/A
**The Netherlands**	**2020**	**Bos et al. [30]**	Retrospective	OPSCC	Postcontrast 3DT1W MRI	In radiomics model 3, 14 combined, 11 unique features overall	Distributed	Shape sphericity, gray level co-occurrence matrix inverse difference moment (Laplacian of Gaussian), kurtosis (wavelet)	Recursive feature elimination, ICC, Man WiU	Logistic regression	Manual	**PyRadiomics 2.2.0**	Not determined	Shape sphericity, gray level co-occurrence matrix inverse difference moment (Laplacian of Gaussian, kurtosis)
Republic of Korea	2020	Y Choi [31]	Retrospective	OPSCC	Contrast-enhanced CT	9	854	original_shape_Flatness original_shape_SphericalDisproportion wavelet_HLH_firstorder_Mean wavelet_HLH_firstorder_Uniformity wavelet_HLH_glcm_ClusterShade wavelet_LHH_glcm_Idm wavelet_LHH_glcm_Imc1 wavelet_LHL_glszm_SmallAreaHighGrayLevelEmphasis wavelet_LLH_glcm_Imc2	Boruta (random forest-based wrapper algorithm)	None	Manual	**PyRadiomics**	syngo.via frontier software (Siemens)	Spatial heterogeneity of tumors might be associated with patients’ prognosis
United States	2020	Stefan P Haider [32]	Retrospective	OPSCC	PET/CT	30, 21, 9	2074	N/A	Various	XGB, SVM, RF, NB, EINET	Manual, separately on PET and CT	**Pyradiomcis**	3D slicer	Radiomics signatures based on primary tumor lesions had higher predictive performance compared to those based on metastatic cervical lymph nodes; however, models using consensus VOI of all nodes had better classification performance than individual nodes
**China**	**2020**	**Ren [33]**	Retrospective	OPSCC	CT	6,7	86	Inverse variance, gray level nonuniformity, inverse variance, energy, dependence variance, informational measure of correlation 1, informational measure of correlation 2, inverse variance, correlation, median, inverse variance, correlation, gray level nonuniformity	ICC, wrapper-based algorithm	k-nearest neighbors (k-NN), logistic regression (LR), and random forest (RF)	Manual	**PyRadiomics 2.2.0**	3D slicer	N/A
**Republic of Korea**	**2020**	**Suh [34]**	Retrospective	OPSCC	T1WI, T2WI, CE-T1WI, and ADC maps	221	1618	Not determined individually	LASSO	Logistic regression, random forest	Manual	**Matlab**	Medical imaging interaction toolkit (MITK)	Features extracted from ADC maps were the most relevant in differentiating HPV-positive and -negative lesions
Italy	2019	Mungai [35]	Retrospective	OPSCC	Contrast-enhanced CT imaging	13	42	LRE, SRHGE, LRLGE, LRHGE, GLNU, coarseness, contrast, busyness, LZE, LZLGE, LZHGE, GLNU	Independent samples *t*-test	N/A	Manual	(***LIFEx***)	Syngo.via, Siemens	N/A
**China**	**2019**	**Huang et al. [36]**	Retrospective	OPSCC	CE-CT	5	540	firstOrder_InterquartileRange, waveletHHH_firstOrder_Kurtosis, waveletLLL_firstOrder_Maximum, shapeSize_Solidity, firstOrder_QuartileCoefficientDispersion	Minimum redundancy maximum relevance (mRMR), LASSO	Logistic regression	Manual	**Matlab 2018a**	Matlab 2018a	N/A
Republic of Korea	2019	Ji Young Lee et al. [37]	Retrospective	OPSCC	Contrast-enhanced CT	N/A	6	Mean, SD, entropy, MPP, skewness, kurtosis	N/A	None	Manual	TexRAD,	TexRAD,	N/A
**The Netherlands**	**2018**	**Leijenar [38]**	Retrospective	OPSCC	Contrast-enhanced CT imaging	165,173	902	Laplacian of Gaussian (3 mm) kurtosis, gray level size zone matrix, low gray level large size emphasis, Laplacian of Gaussian (4 mm) 10th percentile, gray level co-occurrence matrix inverse variance, gray level size zone matrix, small zone emphasis	LASSO	Multivariable logistic regression	Manual	**Matlab 2014a**	N/A	Higher homogeneity in HPV-positive group—lower-contrast uptake
**United States**	**2018**	**Ranjbar [39]**	Retrospective	OPSCC	Contrast-enhanced neck CT	3	77	Median and entropy and GLCM entropy	Independent samples t-test	Diagonal quadratic discriminant analysis	Manual	OsiriX	OsiriX	Difference in texture features could reflect the different amounts of water and keratinization in tumors
Italy	2018	Ravanelli [40]	Retrospective	OPSCC	T1, T2, DWI	8	30	Mean ADC, MPP T2 SSF 4, MPP T2 SSF 3, mean T2 SSF 2, mean T2 SSF 3, mean T2 SSF 1, mean T2 SSF 4, kurtosis VIBE SSF 4	Mann–Whitney U test or Student t-test	N/A	Manual	TexRAD	TexRAD	HPV-positive tumors had a significantly lower ADC than HPV-negative tumors
Switzerland	2017	Bogowicz et al. [41]	Retrospective	Stage III and IV HNSCC	Pre-treatment CE-CT	4	317	1. LLL standard deviation 2. LLL small-zone high gray level emphasis 3. HHL difference entropy 4. Coefficient of variation	Principal component analysis + Horn method	Multivariable Cox regression analysis	Manual	In-house-developed Python	In-house-developed Python	More homogeneity, fewer small regions with high intensity
United States	2017	Yu et al. [42]	Retrospective	OPSCC	CT	2	1683	Mean breadth and spherical disproportion	Wilcoxon rank-sum test, AUC of each future alone	Logistic regression	Manual	**IBEX**	IBEX	HPV-positive patients usually have smaller and simpler tumors
United States	2016	Fujita et al. [43]	Retrospective	None OPSCC HNC	Contrast-enhanced CT	16	42	5 histogram, 3 gray level co-occurrence matrix), 1 gray level run-length feature, 2 gray level gradient matrix features, and 5 law features		None	Manual	**MATLAB**	AW workstation	Spatially dependent variations in features could be used to represent histologic differences in HPV-positive and -negative tumors
United States	2015	Buch et al. [44]	Retrospective	OPSCC	Contrast-enhanced CT	3	42	Entropy, median, GLCM texture feature entropy	Student *t*-test	None	Manual	In-house-developed Matlab	AW workstation (GE Healthcare)	N/A

## Data Availability

All data will be made available upon request to the corresponding author.

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
