# Peer review of "Radiomics Features in Predicting Human Papillomavirus Status in Oropharyngeal Squamous Cell Carcinoma: A Systematic Review, Quality Appraisal, and Meta-Analysis"

_diagnostics, 2024, doi:10.3390/diagnostics14070737_

Round 1

Reviewer 1 Report

Comments and Suggestions for Authors

Dear Authors,

Congratulations on your hard work. Your research effectively addresses an important research question in the field of oncology and radiology by exploring the diagnostic potential of radiomics features in determining HPV status in OPSCC. Enhancements in methodology and inclusion of recent relevant studies could further strengthen the manuscript's impact and credibility.

Here are my suggestions:

1. The methodology employed in this study appears robust, with a comprehensive literature search, systematic review, and quality appraisal. However, there are areas for improvement: external validation of the findings from different datasets could enhance the generalizability of results; utilization of standardized tools (such as Image Biomarker Standardization Initiative - IBSI) for feature extraction and nomenclature may enhance reproducibility and benchmarking; the inclusion of additional studies focusing on different stages of OPSCC and standardized protocols for HPV detection.

2. The Introduction/Discussion section might be expanded with 1-2 parragraphs related to other prognostic/predictive factors in head and neck cancer (e. g. Alexandra G, Alexandru M, Stefan CF, et al. Blood Group Type Association with Head and Neck Cancer. Hematol Rep. 2022;14(1):24-30. Published 2022 Mar 2. doi:10.3390/hematolrep14010005; Kakava K, Karelas I, Koutrafouris I, et al. Relationship between ABO blood groups and head and neck cancer among Greek patients. J BUON. 2016;21(3):594-596.; Singh K, Kote S, Patthi B, et al. Relative Risk of Various Head and Neck Cancers among Different Blood Groups: An Analytical Study. J Clin Diagn Res. 2014;8(4):ZC25-ZC28. doi:10.7860/JCDR/2014/7949.4244; Jaleel BF, Nagarajappa R. Relationship between ABO blood groups and oral cancer. Indian J Dent Res. 2012;23(1):7-10. doi:10.4103/0970-9290.99029)

Author Response

Response to Reviewer Number 1:

Dear Editor of Diagnostics

As the corresponding author I would like to thank you for the opportunity to resubmit our manuscript. We have read these comments with great interest and have made the following changes to the manuscript:

  • Comment number 1:

The methodology employed in this study appears robust, with a comprehensive literature search, systematic review, and quality appraisal. However, there are areas for improvement: external validation of the findings from different datasets could enhance the generalizability of results; utilization of standardized tools (such as Image Biomarker Standardization Initiative - IBSI) for feature extraction and nomenclature may enhance reproducibility and benchmarking; the inclusion of additional studies focusing on different stages of OPSCC and standardized protocols for HPV detection.

Response: Thank you for your comment. We added the requested information to Table2, and added more in this regard to the discussion. We would like to point out that we included all studies that were published until the date the study was submitted, and unfortunately most of the studies did not provide information about the staging and grading of the tumors. We also clarified this issue in the limitations. Furthermore, not all of the studies had external validation performed, we also highlighted this limitation facing the studies in the discussion section.

  • Comment number 2:

The Introduction/Discussion section might be expanded with 1-2 paragraphs related to other prognostic/predictive factors in head and neck cancer (e. g. Alexandra G, Alexandru M, Stefan CF, et al. Blood Group Type Association with Head and Neck Cancer. Hematol Rep. 2022;14(1):24-30. Published 2022 Mar 2. doi:10.7860/JCDR/2014/7949.4244; Jaleel BF, Nagarajappa R. Relationship between ABO blood groups and oral cancer. Indian J Dent Res. 2012;23(1):7-10. doi:10.4103/0970-9290.99029).

Response: Thank you for the comment. We included the above mentioned references in the manuscript, however we refrained from discussing the topic in detail, as none of the studies we included in our analysis considered the relation between blood group and HPV status. However, we acknowledged their importance with adding the references mentioned above to the discussion and introduction of the manuscript.

We thank you for the opportunity to resubmit our manuscript, we also declare that we are ready to undertake any necessary corrections to meet the high standards of the journal.

Sincerely

The corresponding author

Reviewer 2 Report

Comments and Suggestions for Authors

The proposed study was overall very well performed: the subject matter is a current topic, with important insights for the future management of HPV – positive OSCC. Regarding the introduction, it establishes the originality of the research aims by demonstrating the need for investigations in the topic area. It summarizes the recent research related to the topic. However, the research aims should be more explicit and match with the title. It is not clear if the main objective of the paper is to evaluate the accuracy of the radiomics, to compare it with the other imaging methodologies, such as CT or MRI, and/or predicting the features of the HPV – related OSCC observed in radiomics. With regard to materials and methods, the standard guidelines were followed. The statistical analysis performed was clear. The results are reported in a correct manner with the right references to the statistical analysis performed. The significance of the results was underlined. The tables are clear and easy to understand. Nonetheless, it would be appreciated to show the picture of the observed details to provide a reference for the qualitative analysis. The discussion is comprehensive and all the pitfalls of the study were identified in the correct way.  

Author Response

Reviewer Number 1:

Dear Editor of Diagnostics:

As the corresponding author, I would like to thank you for the opportunity to resubmit our manuscript. We have read these comments with great interest and have made the following changes to the manuscript:

The proposed study was overall very well performed: the subject matter is a current topic, with important insights for the future management of HPV – positive OSCC. Regarding the introduction, it establishes the originality of the research aims by demonstrating the need for investigations in the topic area. It summarizes the recent research related to the topic. However, the research aims should be more explicit and match with the title. It is not clear if the main objective of the paper is to evaluate the accuracy of the radiomics, to compare it with the other imaging methodologies, such as CT or MRI, and/or predicting the features of the HPV – related OSCC observed in radiomics. With regard to materials and methods, the standard guidelines were followed. The statistical analysis performed was clear. The results are reported in a correct manner with the right references to the statistical analysis performed. The significance of the results was underlined. The tables are clear and easy to understand. Nonetheless, it would be appreciated to show the picture of the observed details to provide a reference for the qualitative analysis. The discussion is comprehensive and all the pitfalls of the study were identified in the correct way.

Response: Thank you for the comment. We have added more in regards to the objectives of the study in the introduction section. Regarding a picture of observed details, none of the studies we included in our study generated a “feature map” to show the utility of the radiomics features. We agree that the existence of such a map would increase the openness and translatability of the model. We included a statement in the discussion section highlighting the importance of using feature maps and other means of data visualization in increasing the transparency of radiomics models.

We thank you for the opportunity to resubmit our manuscript, we also declare that we are ready to undertake any additional corrections to meet the high standards of the journal.

Sincerely

The corresponding author

Round 2

Reviewer 1 Report

Comments and Suggestions for Authors

Dear Authors,

In the current revised form the manuscript is acceptable for publication.